# DAGs with NO TEARS:
# Continuous Optimization for Structure Learning

**Xun Zheng**[1], **Bryon Aragam**[1], **Pradeep Ravikumar**[1], **Eric P. Xing**[1,2]
[1]Carnegie Mellon University    [2]Petuum Inc.
{xunzheng,naragam,pradeepr,epxing}@cs.cmu.edu

## Abstract

Estimating the structure of directed acyclic graphs (DAGs, also known as Bayesian networks) is a challenging problem since the search space of DAGs is combinatorial and scales superexponentially with the number of nodes. Existing approaches rely on various local heuristics for enforcing the acyclicity constraint. In this paper, we introduce a fundamentally different strategy: we formulate the structure learning problem as a purely *continuous* optimization problem over real matrices that avoids this combinatorial constraint entirely. This is achieved by a novel characterization of acyclicity that is not only smooth but also exact. The resulting problem can be efficiently solved by standard numerical algorithms, which also makes implementation effortless. The proposed method outperforms existing ones, without imposing any structural assumptions on the graph such as bounded treewidth or in-degree.

## 1  Introduction

Learning directed acyclic graphs (DAGs) from data is an NP-hard problem [8, 11], owing mainly to the combinatorial acyclicity constraint that is difficult to enforce efficiently. At the same time, DAGs are popular models in practice, with applications in biology [33], genetics [49], machine learning [22], and causal inference [42]. For this reason, the development of new methods for learning DAGs remains a central challenge in machine learning and statistics.

In this paper, we propose a new approach for score-based learning of DAGs by converting the traditional *combinatorial* optimization problem (left) into a *continuous* program (right):

$$\min_{W \in \mathbb{R}^{d \times d}} F(W) \qquad \min_{W \in \mathbb{R}^{d \times d}} F(W)$$
$$\text{subject to } \mathsf{G}(W) \in \mathsf{DAGs} \quad \Longleftrightarrow \quad \text{subject to } h(W) = 0, \tag{1}$$

where $\mathsf{G}(W)$ is the $d$-node graph induced by the weighted adjacency matrix $W$, $F : \mathbb{R}^{d \times d} \to \mathbb{R}$ is a score function (see Section 2.1 for details), and our key technical device $h : \mathbb{R}^{d \times d} \to \mathbb{R}$ is a smooth function over real matrices, whose level set at zero exactly characterizes acyclic graphs. Although the two problems are equivalent, the continuous program on the right eliminates the need for specialized algorithms that are tailored to search over the combinatorial space of DAGs. Instead, we are able to leverage standard numerical algorithms for constrained problems, which makes implementation particularly easy, not requiring any knowledge about graphical models. This is similar in spirit to the situation for undirected graphical models, in which the formulation of a continuous log-det program [4] sparked a series of remarkable advances in structure learning for undirected graphs (Section 2.2). Unlike undirected models, which can be reduced to a convex program, however, the program (1) is *nonconvex*. Nonetheless, as we will show, even naïve solutions to this program yield state-of-the-art results for learning DAGs.

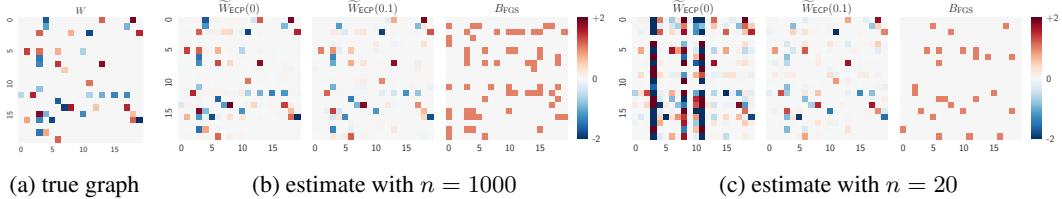

(a) true graph  (b) estimate with $n = 1000$  (c) estimate with $n = 20$

Figure 1: Visual comparison of the learned weighted adjacency matrix on a 20-node graph with $n = 1000$ (large samples) and $n = 20$ (insufficient samples): $\widetilde{W}_{\text{ECP}}(\lambda)$ is the proposed NOTEARS algorithm with $\ell_1$-regularization $\lambda$, and $B_{\text{FGS}}$ is the binary estimate of the baseline [31]. The proposed algorithms perform well on large samples, and remains accurate on small $n$ with $\ell_1$ regularization.

**Contributions.**   The main thrust of this work is to re-formulate score-based learning of DAGs so that standard smooth optimization schemes such as L-BFGS [28] can be leveraged. To accomplish this, we make the following specific contributions:

- We explicitly construct a smooth function over $\mathbb{R}^{d \times d}$ with computable derivatives that encodes the acyclicity constraint. This allows us to replace the combinatorial constraint $\mathsf{G} \in \mathbb{D}$ in (4) with a smooth equality constraint.

- We develop an equality-constrained program for simultaneously estimating the structure and parameters of a sparse DAG from possibly high-dimensional data, and show how standard numerical solvers can be used to find stationary points.

- We demonstrate the effectiveness of the resulting method in empirical evaluations against existing state-of-the-arts. See Figure 1 for a quick illustration and Section 5 for details.

- We compare our ouput to the exact global minimizer [12], and show that our method attains scores that are comparable to the globally optimal score in practice, although our methods are only guaranteed to find stationary points.

Most interestingly, our approach is very simple and can be implemented in about 50 lines of Python code. As a result of its simplicity and effortlessness in its implementation, we call the resulting method NOTEARS: *Non-combinatorial Optimization via Trace Exponential and Augmented lagRangian for Structure learning.* The implementation is publicly available at `https://github.com/xunzheng/notears`.

## 2 Background

The basic DAG learning problem is formulated as follows: Let $\mathbf{X} \in \mathbb{R}^{n \times d}$ be a data matrix consisting of $n$ i.i.d. observations of the random vector $X = (X_1, \ldots, X_d)$ and let $\mathbb{D}$ denote the (discrete) space of DAGs $\mathsf{G} = (\mathsf{V}, \mathsf{E})$ on $d$ nodes. Given $\mathbf{X}$, we seek to learn a DAG $\mathsf{G} \in \mathbb{D}$ (also called a *Bayesian network*) for the joint distribution $\mathbb{P}(X)$ [22, 42]. We model $X$ via a structural equation model (SEM) defined by a weighted adjacency matrix $W \in \mathbb{R}^{d \times d}$. Thus, instead of operating on the discrete space $\mathbb{D}$, we will operate on $\mathbb{R}^{d \times d}$, the continuous space of $d \times d$ real matrices.

### 2.1 Score functions and SEM

Any $W \in \mathbb{R}^{d \times d}$ defines a graph on $d$ nodes in the following way: Let $\mathcal{A}(W) \in \{0, 1\}^{d \times d}$ be the binary matrix such that $[\mathcal{A}(W)]_{ij} = 1 \iff w_{ij} \neq 0$ and zero otherwise; then $\mathcal{A}(W)$ defines the adjacency matrix of a directed graph $\mathsf{G}(W)$. In a slight abuse of notation, we will thus treat $W$ as if it were a (weighted) graph. In addition to the graph $\mathsf{G}(W)$, $W = [\, w_1 \,|\, \cdots \,|\, w_d \,]$ defines a linear SEM by $X_j = w_j^T X + z_j$, where $X = (X_1, \ldots, X_d)$ is a random vector and $z = (z_1, \ldots, z_d)$ is a random noise vector. We do *not* assume that $z$ is Gaussian. More generally, we can model $X_j$ via a generalized linear model (GLM) $\mathbb{E}(X_j \,|\, X_{\text{pa}(X_j)}) = f(w_j^T X)$. For example, if $X_j \in \{0, 1\}$, we can model the conditional distribution of $X_j$ given its parents via logistic regression.

In this paper, we focus on linear SEM and the least-squares (LS) loss $\ell(W; \mathbf{X}) = \frac{1}{2n}\|\mathbf{X} - \mathbf{X}W\|_F^2$, although everything in the sequel applies to any smooth loss function $\ell$ defined over $\mathbb{R}^{d \times d}$. The

statistical properties of the LS loss in scoring DAGs have been extensively studied: The minimizer of the LS loss provably recovers a true DAG with high probability on finite-samples and in high-dimensions ($d \gg n$), and hence is consistent for both Gaussian SEM [3, 45] and non-Gaussian SEM [24].[1] Note also that these results imply that the faithfulness assumption is not required in this set-up. Given this extensive previous work on statistical issues, our focus in this paper is entirely on the computational problem of finding an SEM that minimizes the LS loss.

This translation between graphs and SEM is central to our approach. Since we are interested in learning a *sparse* DAG, we add $\ell_1$-regularization $\|W\|_1 = \|\operatorname{vec}(W)\|_1$ resulting in the regularized score function

$$F(W) = \ell(W; \mathbf{X}) + \lambda\|W\|_1 = \frac{1}{2n}\|\mathbf{X} - \mathbf{X}W\|_F^2 + \lambda\|W\|_1. \tag{2}$$

Thus we seek to solve

$$\begin{aligned} \min_{W \in \mathbb{R}^{d \times d}} \quad & F(W) \\ \text{subject to} \quad & \mathsf{G}(W) \in \mathbb{D}. \end{aligned} \tag{3}$$

Unfortunately, although $F(W)$ is continuous, the DAG constraint $\mathsf{G}(W) \in \mathbb{D}$ remains a challenge to enforce. In Section 3, we show how this discrete constraint can be replaced by a smooth equality constraint.

## 2.2 Previous work

Traditionally, score-based learning seeks to optimize a *discrete score* $Q : \mathbb{D} \to \mathbb{R}$ over the set of DAGs $\mathbb{D}$; note that this is distinct from our score $F(W)$ whose domain is $\mathbb{R}^{d \times d}$ instead of $\mathbb{D}$. This can be written as the following combinatorial optimization problem:

$$\begin{aligned} \min_{\mathsf{G}} \quad & Q(\mathsf{G}) \\ \text{subject to} \quad & \mathsf{G} \in \mathbb{D} \end{aligned} \tag{4}$$

Popular score functions include BDe(u) [20], BGe [23], BIC [10], and MDL [6]. Unfortunately, (4) is NP-hard to solve [8, 11] owing mainly to the nonconvex, combinatorial nature of the optimization problem. This is the main drawback of existing approaches for solving (4): The acyclicity constraint is a combinatorial constraint with the number of acyclic structures increasing superexponentially in $d$ [32]. Notwithstanding, there are algorithms for solving (4) to global optimality for small problems [12, 13, 29, 39, 40, 47]. There is also a wide literature on approximate algorithms based on order search [30, 34–36, 43], greedy search [9, 20, 31], and coordinate descent [2, 16, 18]. By searching over the space of topological orderings, the former order-based methods trade-off the difficult problem of enforcing acyclicity with a search over $d!$ orderings, whereas the latter methods enforce acyclicity one edge at a time, explicitly checking for acyclicity violations each time an edge is added. Other approaches that avoid optimizing (4) directly include constraint-based methods [41, 42], hybrid methods [17, 44], and Bayesian methods [14, 27, 51].

It is instructive to compare this problem to a similar and well-understood problem: Learning an undirected graph (Markov network) from data. Score-based methods based on discrete scores similar to (4) proliferated in the early days for learning undirected graphs [e.g. 22, §20.7]. More recently, the re-formulation of this problem as a convex program over real, symmetric matrices [4, 48] has led to extremely efficient algorithms for learning undirected graphs [15, 21, 37]. One of the key factors in this success was having a closed-form, tractable program for which existing techniques from the extensive optimization literature could be applied. Unfortunately, the general problem of DAG learning has not benefitted in this way, arguably due to the intractable form of the program (4). One of our main goals in the current work is to formulate score-based learning via a similar closed-form, continuous program. The key device in accomplishing this is a smooth characterization of acyclicity that will be introduced in the next section.

## 3 A new characterization of acyclicity

In order to make (3) amenable to black-box optimization, we propose to replace the combinatorial acyclicity constraint $\mathsf{G}(W) \in \mathbb{D}$ in (3) with a single smooth equality constraint $h(W) = 0$. Ideally, we would like a function $h : \mathbb{R}^{d \times d} \to \mathbb{R}$ that satisfies the following desiderata:

(a) $h(W) = 0$ if and only if $W$ is acyclic (i.e. $\mathsf{G}(W) \in \mathbb{D}$);
(b) The values of $h$ quantify the "DAG-ness" of the graph;
(c) $h$ is smooth;
(d) $h$ and its derivatives are easy to compute.

Property (b) is useful in practice for diagnostics. By "DAG-ness", we mean some quantification of how severe violations from acyclicity become as $W$ moves further from $\mathbb{D}$. Although there are many ways to satisfy (b) by measuring some notion of "distance" to $\mathbb{D}$, typical approaches would violate (c) and (d). For example, $h$ might be the minimum $\ell_2$ distance to $\mathbb{D}$ or it might be the sum of edge weights along all cyclic paths of $W$, however, these are either non-smooth (violating (c)) or hard to compute (violating (d)). If a function that satisfies desiderata (a)-(d) exists, we can hope to apply existing machinery for constrained optimization such as Lagrange multipliers. Consequently, the DAG learning problem becomes equivalent to solving a numerical optimization problem, which is agnostic about the graph structure.

Our main result establishes the existence of such a function:

**Theorem 1.** *A matrix* $W \in \mathbb{R}^{d \times d}$ *is a DAG if and only if*

$$h(W) = \text{tr}\left(e^{W \circ W}\right) - d = 0, \tag{5}$$

*where* $\circ$ *is the Hadamard product and* $e^A$ *is the matrix exponential of A. Moreover,* $h(W)$ *has a simple gradient*

$$\nabla h(W) = \left(e^{W \circ W}\right)^T \circ 2W, \tag{6}$$

*and satisfies all of the desiderata (a)-(d).*

We sketch a proof of the first claim here; a formal proof of Theorem 1 can be found in Appendix A. Let $S = W \circ W$, then $S \in \mathbb{R}_+^{d \times d}$ while preserving the sparsity pattern of $W$. Recall for any positive integer $k$, the entries of matrix power $(S^k)_{ij}$ is the sum of weight products along all $k$-step paths from node $i$ to node $j$. Since $S$ is nonnegative, $\text{tr}(S^k) = 0$ iff there is no $k$-cycles in the graph. Expanding the power series,

$$\text{tr}(e^S) = \text{tr}(I) + \text{tr}(S) + \frac{1}{2!}\text{tr}(S^2) + \cdots \geq d, \tag{7}$$

and the equality is attained iff the underlying graph of $S$, equivalently $W$, has no cycles.

A key conclusion from Theorem 1 is that $h$ and its gradient only involve evaluating the matrix exponential, which is a well-studied function in numerical anlaysis, and whose $O(d^3)$ algorithm [1] is readily available in many scientific computing libraries. Although the connection between trace of matrix power and number of cycles in the graph is well-known [19], to the best of our knowledge, this characterization of acyclicity has not appeared in the DAG learning literature previously. We defer the discussion of other possible characterizations in the appendix. In the next section, we apply Theorem 1 to solve the program (3) to stationarity by treating it as an equality constrained program.

## 4 Optimization

Theorem 1 establishes a smooth, algebraic characterization of acyclicity that is also computable. As a consequence, the following equality-constrained program ($\mathsf{ECP}$) is equivalent to (3):

$$(\mathsf{ECP}) \qquad \begin{array}{cc} \min_{W \in \mathbb{R}^{d \times d}} & F(W) \\ \text{subject to} & h(W) = 0. \end{array} \tag{8}$$

---

**Algorithm 1** NOTEARS algorithm

---

1. Input: Initial guess $(W_0, \alpha_0)$, progress rate $c \in (0, 1)$, tolerance $\epsilon > 0$, threshold $\omega > 0$.
2. For $t = 0, 1, 2, \dots$ :
    (a) Solve primal $W_{t+1} \leftarrow \arg\min_W L^\rho(W, \alpha_t)$ with $\rho$ such that $h(W_{t+1}) < ch(W_t)$.
    (b) Dual ascent $\alpha_{t+1} \leftarrow \alpha_t + \rho h(W_{t+1})$.
    (c) If $h(W_{t+1}) < \epsilon$, set $\widetilde{W}_{\mathsf{ECP}} = W_{t+1}$ and break.
3. Return the thresholded matrix $\widehat{W} := \widetilde{W}_{\mathsf{ECP}} \circ 1(|\widetilde{W}_{\mathsf{ECP}}| > \omega)$.

---

The main advantage of (ECP) compared to both (3) and (4) is its amenability to classical techniques from the mathematical optimization literature. Nonetheless, since $\{W : h(W) = 0\}$ is a nonconvex constraint, (8) is a nonconvex program, hence we still inherit the difficulties associated with nonconvex optimization. In particular, we will be content to find stationary points of (8); in Section 5.3 we compare our results to the global minimizer and show that the stationary points found by our method are close to global minima in practice.

In the follows, we outline the algorithm for solving (8). It consists of three steps: (i) converting the *constrained* problem into a sequence of *unconstrained* subproblems, (ii) optimizing the unconstrained subproblems, and (iii) thresholding. The full algorithm is outlined in Algorithm 1.

### 4.1 Solving the ECP with augmented Lagrangian

We will use the augmented Lagrangian method [e.g. 25] to solve (ECP), which solves the original problem augmented by a quadratic penalty:

$$
\begin{aligned}
\min_{W \in \mathbb{R}^{d \times d}} \quad & F(W) + \frac{\rho}{2}|h(W)|^2 \\
\text{subject to} \quad & h(W) = 0
\end{aligned}
\tag{9}
$$

with a penalty parameter $\rho > 0$. A nice property of the augmented Lagrangian method is that it approximates well the solution of a *constrained* problem by the solution of *unconstrained* problems *without* increasing the penalty parameter $\rho$ to infinity [25]. The algorithm is essentially a dual ascent method for (9). To begin with, the dual function with Lagrange multiplier $\alpha$ is given by

$$
D(\alpha) = \min_{W \in \mathbb{R}^{d \times d}} L^\rho(W, \alpha),
\tag{10}
$$

$$
\text{where} \quad L^\rho(W, \alpha) = F(W) + \frac{\rho}{2}|h(W)|^2 + \alpha h(W)
\tag{11}
$$

is the augmented Lagrangian. The goal is to find a local solution to the dual problem

$$
\max_{\alpha \in \mathbb{R}} \; D(\alpha).
\tag{12}
$$

Let $W_\alpha^\star$ be the local minimizer of the Lagrangian (10) at $\alpha$, i.e. $D(\alpha) = L^\rho(W_\alpha^\star, \alpha)$. Since the dual objective $D(\alpha)$ is linear in $\alpha$, the derivative is simply given by $\nabla D(\alpha) = h(W_\alpha^\star)$. Therefore one can perform dual gradient ascent to optimize (12):

$$
\alpha \leftarrow \alpha + \rho h(W_\alpha^\star),
\tag{13}
$$

where the choice of step size $\rho$ comes with the following convergence rate:

**Proposition 1** (Corollary 11.2.1, 25)**.** *For $\rho$ large enough and the starting point $\alpha_0$ near the solution $\alpha^\star$, the update* (13) *converges to $\alpha^\star$ linearly.*

In our experiments, typically fewer than 10 steps of the augmented Lagrangian scheme are required.

### 4.2 Solving the unconstrained subproblem

The augmented Lagrangian converts a *constrained* problem (9) into a sequence of *unconstrained* problems (10). We now discuss how to solve these subproblems efficiently. Let $\boldsymbol{w} = \operatorname{vec}(W) \in \mathbb{R}^p$,

with $p = d^2$. The unconstrained subproblem (10) can be considered as a typical minimization problem over real vectors:

$$\min_{\boldsymbol{w} \in \mathbb{R}^p} f(\boldsymbol{w}) + \lambda \|\boldsymbol{w}\|_1, \tag{14}$$

$$\text{where} \quad f(\boldsymbol{w}) = \ell(W; \mathbf{X}) + \frac{\rho}{2}|h(W)|^2 + \alpha h(W) \tag{15}$$

is the smooth part of the objective. Our goal is to solve the above problem to high accuracy so that $h(W)$ can be sufficiently suppressed.

In the special case of $\lambda = 0$, the nonsmooth term vanishes and the problem simply becomes an unconstrained smooth minimization, for which a number of efficient numerical algorithms are available, for instance the L-BFGS [7]. To handle the nonconvexity, a slight modification [28, Procedure 18.2] needs to be applied.

When $\lambda > 0$, the problem becomes composite minimization, which can also be efficiently solved by the proximal quasi-Newton (PQN) method [50]. At each step $k$, the key idea is to find the descent direction through a quadratic approximation of the smooth term:

$$\boldsymbol{d}_k = \arg\min_{\boldsymbol{d} \in \mathbb{R}^p} \boldsymbol{g}_k^T \boldsymbol{d} + \frac{1}{2}\boldsymbol{d}^T B_k \boldsymbol{d} + \lambda \|\boldsymbol{w}_k + \boldsymbol{d}\|_1, \tag{16}$$

where $\boldsymbol{g}_k$ is the gradient of $f(\boldsymbol{w})$ and $B_k$ is the L-BFGS approximation of the Hessian. Note that for each coordinate $j$, problem (16) has a closed form update $\boldsymbol{d} \leftarrow \boldsymbol{d} + z^\star e_j$ given by

$$z^\star = \arg\min_z \frac{1}{2} \underbrace{B_{jj}}_{a} z^2 + (\underbrace{\boldsymbol{g}_j + (B\boldsymbol{d})_j}_{b})z + \lambda|\underbrace{\boldsymbol{w}_j + \boldsymbol{d}_j}_{c} + z| = -c + S\left(c - \frac{b}{a}, \frac{\lambda}{a}\right). \tag{17}$$

Moreover, the low-rank structure of $B_k$ enables fast computation for coordinate update. As we describe in Appendix B, the precomputation time is only $O(m^2 p + m^3)$ where $m \ll p$ is the memory size of L-BFGS, and each coordinate update is $O(m)$. Furthermore, since we are using sparsity regularization, we can further speed up the algorithm by aggressively shrinking the active set of coordinates based on their subgradients [50], and exclude the remaining dimensions from being updated. With the updates restricted to the active set $\mathcal{S}$, all dependencies of the complexity on $O(p)$ becomes $O(|\mathcal{S}|)$, which is substantially smaller. Hence the overall complexity of L-BFGS update is $O(m^2|\mathcal{S}| + m^3 + m|\mathcal{S}|T)$, where $T$ is the number of inner iterations, typically $T = 10$.

### 4.3 Thresholding

In regression problems, it is known that post-processing estimates of coefficients via hard thresholding provably reduces the number of false discoveries [46, 52]. Motivated by these encouraging results, we threshold the edge weights as follows: After obtaining a stationary point $\widetilde{W}_{\mathsf{ECP}}$ of (9), given a fixed threshold $\omega > 0$, set any weights smaller than $\omega$ in absolute value to zero. This strategy also has the important effect of "rounding" the numerical solution of the augmented Lagrangian (9), since due to numerical precisions the solution satisfies $h(\widetilde{W}_{\mathsf{ECP}}) \leq \epsilon$ for some small tolerance $\epsilon$ near machine precision (e.g. $\epsilon = 10^{-8}$), rather than $h(\widetilde{W}_{\mathsf{ECP}}) = 0$ strictly. However, since $h(\widetilde{W}_{\mathsf{ECP}})$ explicitly *quantifies* the "DAG-ness" of $\widetilde{W}_{\mathsf{ECP}}$ (see desiderata (b), Section 3), a small threshold $\omega$ suffices to rule out cycle-inducing edges.

## 5 Experiments

We compared our method against greedy equivalent search (GES) [9, 31], the PC algorithm [42], and LiNGAM [38]. For GES, we used the fast greedy search (FGS) implementation from Ramsey *et al.* [31]. Since the accuracy of PC and LiNGAM was significantly lower than either FGS or NOTEARS, we only report the results against FGS here. This is consistent with previous work on score-based learning [2], which also indicates that FGS outperforms other techniques such as hill-climbing and MMHC [44]. FGS was chosen since it is a state-of-the-art algorithm that scales to large problems.

For brevity, we outline the basic set-up of our experiments here; precise details of our experimental set-up, including all parameter choices and more detailed evaluations, can be found in Appendix E.

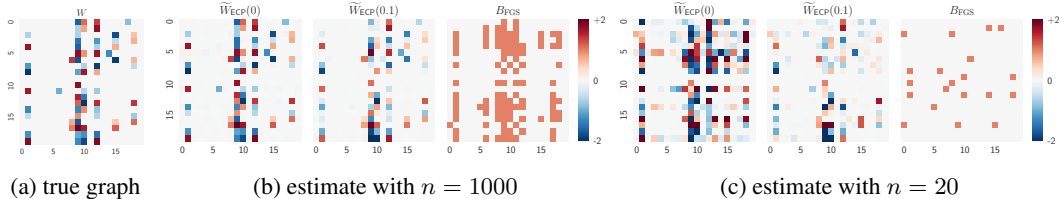

(a) true graph         (b) estimate with $n = 1000$        (c) estimate with $n = 20$

Figure 2: Parameter estimates of $\widetilde{W}_{\text{ECP}}$ on a scale-free graph. Without the additional thresholding step in Algorithm 1, NOTEARS still produces consistent estimates of the true graph. The proposed method estimates the weights very well with large samples even without regularization, and remains accurate on insufficient samples when $\ell_1$-regularization is introduced. See also Figure 1.

In each experiment, a random graph G was generated from one of two random graph models, Erdös-Rényi (ER) or scale-free (SF). Given G, we assigned uniformly random edge weights to obtain a weight matrix $W$. Given $W$, we sampled $X = W^T X + z \in \mathbb{R}^d$ from three different noise models: Gaussian (Gauss), Exponential (Exp), and Gumbel (Gumbel). Based on these models, we generated random datasets $\mathbf{X} \in \mathbb{R}^{n \times d}$ by generating rows i.i.d. according to one of these three models with $d \in \{10, 20, 50, 100\}$ and $n \in \{20, 1000\}$. Since FGS outputs a CPDAG instead of a DAG or weight matrix, some care needs to be taken in making comparisons; see Appendix E.1 for details.

## 5.1 Parameter estimation

We first performed a qualitative study of the solutions obtained by NOTEARS *without thresholding* by visualizing the weight matrix $\widetilde{W}_{\text{ECP}}$ obtained by solving (ECP) (i.e. $\omega = 0$). This is illustrated in Figures 1 (ER-2) and 2 (SF-4). The key takeaway is that our method provides (empirically) consistent parameter estimates of the true weight matrix $W$. The final thresholding step in Algorithm 1 is only needed to ensure accuracy in structure learning. It also shows how effective is $\ell_1$-regularization in small $n$ regime.

## 5.2 Structure learning

We now examine our method for structure recovery, which is shown in Figure 3. For brevity, we only report the numbers for the structural Hamming distance (SHD) here, but complete figures and tables for additional metrics can be found in the supplement. Consistent with previous work on greedy methods, FGS is very competitive when the number of edges is small (ER-2), but rapidly deteriorates for even modest numbers of edges (SF-4). In the latter regime, NOTEARS shows significant improvements. This is consistent across each metric we evaluated, and the difference grows as the number of nodes $d$ gets larger. Also notice that our algorithm performs uniformly better for each noise model (Exp, Gauss, and Gumbel), without leveraging any specific knowledge about the noise type. Again, $\ell_1$-regularizer helps significantly in the small $n$ setting.

## 5.3 Comparison to exact global minimizer

In order to assess the ability of our method to solve the original program given by (3), we used the GOBNILP program [12, 13] to find the exact minimizer of (3). Since this involves enumerating all possible parent sets for each node, these experiments are limited to small DAGs. Nonetheless, these small-scale experiments yield valuable insight into how well NOTEARS performs in actually solving the original problem. In our experiments we generated random graphs with $d = 10$, and then generated 10 simulated datasets containing $n = 20$ samples (for high-dimensions) and $n = 1000$ (for low-dimensions). We then compared the scores returned by our method to the exact global minimizer computed by GOBNILP along with the estimated parameters. The results are shown in Table 1. Surprisingly, although NOTEARS is only guaranteed to return a local minimizer, in many cases the obtained solution is very close to the global minimizer, as evidenced by deviations $\|\widehat{W} - W_{\text{G}}\|$. Since the general structure learning problem is NP-hard, we suspect that although the models we have tested (i.e. ER and SF) appear amenable to fast solution, in the worst-case there are graphs which will still take exponential time to run or get stuck in a local minimum. Furthermore, the problem becomes

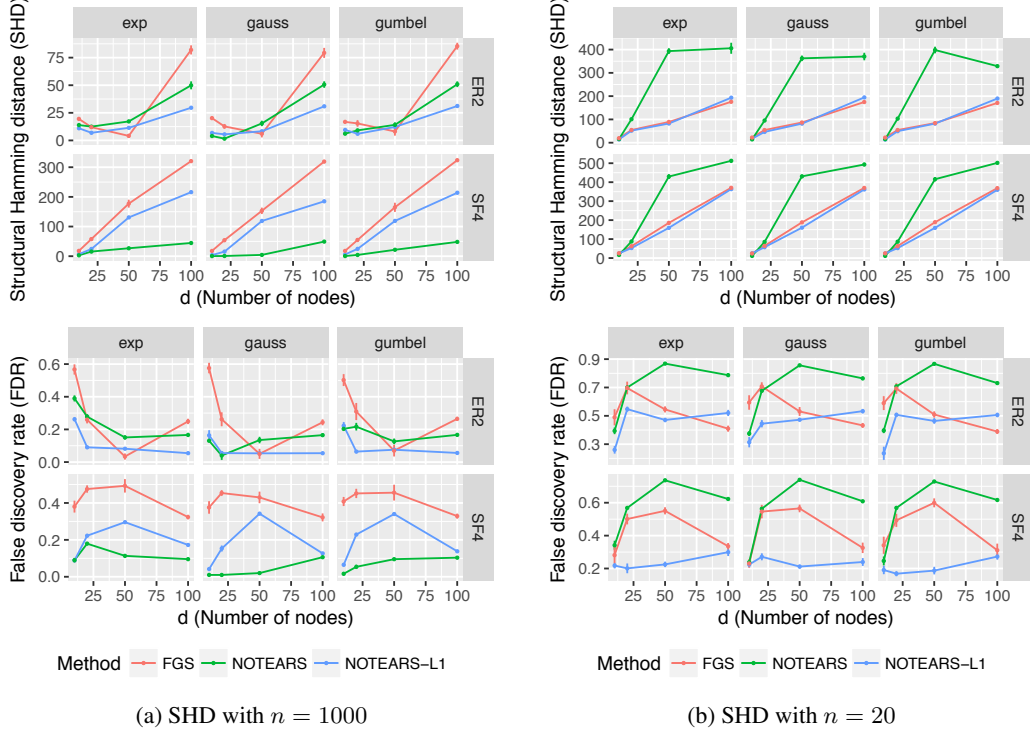

(a) SHD with $n = 1000$

(b) SHD with $n = 20$

Figure 3: Structure recovery in terms of SHD and FDR to the true graph (lower is better). Rows: random graph types, {ER,SF}-$k$ = {Erdös-Rényi, scale-free} graphs with $kd$ expected edges. Columns: noise types of SEM. Error bars represent standard errors over 10 simulations.

Table 1: Comparison of NOTEARS vs. globally optimal solution. $\Delta(W_{\mathsf{G}}, \widehat{W}) = F(W_{\mathsf{G}}) - F(\widehat{W})$.

| $n$ | $\lambda$ | Graph | $F(W)$ | $F(W_{\mathsf{G}})$ | $F(\widehat{W})$ | $F(\widetilde{W}_{\mathsf{ECP}})$ | $\Delta(W_{\mathsf{G}}, \widehat{W})$ | $\|\widehat{W} - W_{\mathsf{G}}\|$ | $\|W - W_{\mathsf{G}}\|$ |
|---|---|---|---|---|---|---|---|---|---|
| 20 | 0 | ER2 | 5.11 | 3.85 | 5.36 | 3.88 | -1.52 | 0.07 | 3.38 |
| 20 | 0.5 | ER2 | 16.04 | 12.81 | 13.49 | 12.90 | -0.68 | 0.12 | 3.15 |
| 1000 | 0 | ER2 | 4.99 | 4.97 | 5.02 | 4.95 | -0.05 | 0.02 | 0.40 |
| 1000 | 0.5 | ER2 | 15.93 | 13.32 | 14.03 | 13.46 | -0.71 | 0.12 | 2.95 |
| 20 | 0 | SF4 | 4.99 | 3.77 | 4.70 | 3.85 | -0.93 | 0.08 | 3.31 |
| 20 | 0.5 | SF4 | 23.33 | 16.19 | 17.31 | 16.69 | -1.12 | 0.15 | 5.08 |
| 1000 | 0 | SF4 | 4.96 | 4.94 | 5.05 | 4.99 | -0.11 | 0.04 | 0.29 |
| 1000 | 0.5 | SF4 | 23.29 | 17.56 | 19.70 | 18.43 | -2.13 | 0.13 | 4.34 |

more difficult as $d$ increases. Nonetheless, this is encouraging evidence that the nonconvexity of (8) is a minor issue in practice. We leave it to future work to investigate these problems further.

## 5.4 Real-data

We also compared FGS and NOTEARS on a real dataset provided by Sachs *et al.* [33]. This dataset consists of continuous measurements of expression levels of proteins and phospholipids in human immune system cells ($n = 7466$ $d = 11$, 20 edges). This dataset is a common benchmark in graphical models since it comes with a known *consensus network*, that is, a gold standard network based on experimental annotations that is widely accepted by the biological community. In our experiments, FGS estimated 17 total edges with an SHD of 22, compared to 16 for NOTEARS with an SHD of 22.

# 6 Discussion

We have proposed a new method for learning DAGs from data based on a continuous optimization program. This represents a significant departure from existing approaches that search over the discrete space of DAGs, resulting in a difficult optimization program. We also proposed two optimization schemes for solving the resulting program to stationarity, and illustrated its advantages over existing methods such as greedy equivalence search. Crucially, by performing global updates (e.g. all parameters at once) instead of local updates (e.g. one edge at a time) in each iteration, our method is able to avoid relying on assumptions about the local structure of the graph. To conclude, let us discuss some of the limitations of our method and possible directions for future work.

First, it is worth emphasizing once more that the equality constrained program (8) is a nonconvex program. Thus, although we overcome the difficulties of *combinatorial* optimization, our formulation still inherits the difficulties associated with *nonconvex* optimization. In particular, black-box solvers can at best find stationary points of (8). With the exception of exact methods, however, existing methods suffer from this drawback as well.[2] The main advantage of NOTEARS then is *smooth*, *global search*, as opposed to combinatorial, local search; and furthermore the search is delegated to standard numerical solvers.

Second, the current work relies on the smoothness of the score function, in order to make use of gradient-based numerical solvers to guide the graph search. However it is also interesting to consider non-smooth, even discrete scores such as BDe [20]. Off-the-shelf techniques such as Nesterov's smoothing [26] could be useful, however more thorough investigation is left for future work.

Third, since the evaluation of the matrix exponential is $O(d^3)$, the computational complexity of our method is cubic in the number of nodes, although the constant is small for sparse matrices. In fact, this is one of the key motivations for our use of second-order methods (as opposed to first-order), i.e. to reduce the number of matrix exponential computations. By using second-order methods, each iteration make significantly more progress than first-order methods. Furthermore, although in practice not many iterations ($t \sim 10$) are required, we have not established any worst-case iteration complexity results. In light of the results in Section 5.3, we expect there are exceptional cases where convergence is slow. Notwithstanding, NOTEARS already outperforms existing methods when the in-degree is large, which is known difficult spot for existing methods. We leave it to future work to study these cases in more depth.

Lastly, in our experiments, we chose a fixed, suboptimal value of $\omega > 0$ for thresholding (Section 4.3). Clearly, it would be preferable to find a data-driven choice of $\omega$ that adapts to different noise-to-signal ratios and graph types. It is an intersting direction for future to study such choices.

The code is publicly available at `https://github.com/xunzheng/notears`.

**Acknowledgments**

We thank the anonymous reviewers for valuable feedback. P.R. acknowledges the support of NSF via IIS-1149803, IIS-1664720. E.X. and B.A. acknowledge the support of NIH R01GM114311, P30DA035778. X.Z. acknowledges the support of Dept of Health BD4BH4100070287, NSF IIS1563887, AFRL/DARPA FA87501720152.

## Footnotes

[1]Due to nonconvexity, there may be more than one minimizer: These and other technical issues such as parameter identifiability are addressed in detail in the cited references.

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
