[Supplementary Material]

# A   Proof of Theorem 1

We proceed in two steps: First, we consider the simpler case of binary adjacency matrices $B \in \{0,1\}^{d \times d}$ (Section A.1). Note that since $\{0,1\}^{d \times d}$ is a discrete space, we cannot take gradients or do continuous optimization. For this we need the second step, in which we relax the function we originally define on binary matrices to real matrices (Section A.2).

## A.1   Special case: Binary adjacency matrices

When does a matrix $B \in \{0,1\}^{d \times d}$ correspond to an acyclic graph? Recall the *spectral radius* $r(B)$ of a matrix $B$ is the largest absolute eigenvalue of $B$. One simple characterization of acyclicity is the following:

**Proposition 2** (Infinite series)**.** *Suppose $B \in \{0,1\}^{d \times d}$ and $r(B) < 1$. Then $B$ is a DAG if and only if*

$$\mathrm{tr}(I - B)^{-1} = d. \tag{18}$$

*Proof.* It essentially boils down to the fact that $\mathrm{tr}\, B^k$ counts the number of length-$k$ closed walks in a directed graph. Clearly an acyclic graph will have $\mathrm{tr}\, B^k = 0$ for all $k = 1, \ldots, \infty$. In other words, $B$ has no cycles if and only if $f(B) = \sum_{k=1}^{\infty} \sum_{i=1}^{d} (B^k)_{ii} = 0$, then

$$\mathrm{tr}(I - B)^{-1} = \mathrm{tr} \sum_{k=0}^{\infty} B^k = \mathrm{tr}\, I + \sum_{k=1}^{\infty} \mathrm{tr}\, B^k = d + \sum_{k=1}^{\infty} \sum_{i=1}^{d} (B^k)_{ii} = d + f(B).$$

The desired result follows.  □

Unfortunately, the condition that $r(B) < 1$ is strong: although it is automatically satisfied when $B$ is a DAG, it is generally not true otherwise, and furthermore the projection is nontrivial. Alternatively, instead of the infinite series, one could consider the characterization based on *finite* series $\sum_{k=1}^{d} \mathrm{tr}\, B^k = 0$, which does not require $r(B) < 1$. However, this is impractical for numerical reasons: The entries of $B^k$ can easily exceed machine precision for even small values of $d$, which makes both function and gradient evaluations highly unstable. Therefore it remains to find a characterization that not only holds for all possible $B$, but also has numerical stability.

**Proposition 3** (Matrix exponential)**.** *A binary matrix $B \in \{0,1\}^{d \times d}$ is a DAG if and only if*

$$\mathrm{tr}\, e^B = d. \tag{19}$$

*Proof.* Similar to Proposition 2 by noting that $B$ has no cycles if and only if $(B^k)_{ii} = 0$ for all $k \geq 1$ and all $i$, which is true if and only if $\sum_{k=1}^{\infty} \sum_{i=1}^{d} (B^k)_{ii}/k! = \mathrm{tr}\, e^B - d = 0$.  □

It is worth pointing out that matrix exponential is well-defined for all square matrices. In addition to everywhere convergence, this characterization has an added bonus: As the number of edges in $B$ increases along with the number of nodes $d$, the number of possible closed walks grows rapidly, so the trace characterization $\mathrm{tr}(I - B)^{-1}$ rapidly becomes ill-conditioned and difficult to manage. By re-weighting the number of length-$k$ closed walks by $k!$, this becomes much easier to manage. While this is a useful characterization, it does not satisfy all of our desiderata since—being defined over a discrete space—it is not a smooth function. The final step is to extend Proposition 3 to all of $\mathbb{R}^{d \times d}$.

## A.2   The general case: Weighted adjacency matrices

Unfortunately, the characterization (19) fails if we replace $B$ with an arbitrary weighted matrix $W$. However, we can replace $B$ with any *nonnegative* weighted matrix, and the same argument use to prove Proposition 3 shows that (19) will still characterize acyclicity. Thus, to extend this to matrices with both positive and negative values, we can simply use the Hadamard product $W \circ W$, which leads to

$$h(W) = \mathrm{tr}\left(e^{W \circ W}\right) - d = 0 \tag{20}$$

and its gradient

$$\nabla h(W) = \left(e^{W \circ W}\right)^T \circ 2W. \tag{21}$$

The proof of (5) is similar to (19), and desiderata (c)-(d) follow from (6). To see why desiderata (b) holds, note that the proof of Proposition 2 shows that the power series $\mathrm{tr}(B + B^2 + \cdots)$ simply counts the number of closed walks in $B$, and the matrix exponential simply re-weights these counts. Replacing $B$ with $W \circ W$ amounts to counting *weighted* closed walks, where the weight of each edge is $w_{ij}^2$. Thus, larger $h(W) > h(W')$ means either (a) $W$ has more cycles than $W'$ or (b) The cycles in $W$ are more heavily weighted than in $W'$.

Moreover, notice that $h(W) \geq 0$ for all $W$ since each term in the series is nonnegative. This gives another interesting perspective of the space of DAGs as the set of global minima of $h(W)$. However, due to the nonconvexity, this is not equivalent to the first order stationary condition $\nabla h(W) = 0$.

## B  Details of Proximal Quasi-Newton

Recall $B_k \in \mathbb{R}^{p \times p}$ is the low-rank approximation of the Hessian matrix given by L-BFGS updates. Let the memory size of L-BFGS be $m$, which is taken to be $m \ll p$. The compact form of L-BFGS update can be written as

$$B_k = \gamma_k I - Q\widehat{Q}, \tag{22}$$

where

$$Q = \begin{bmatrix} \gamma_k S_k & Y_k \end{bmatrix}, \ R = \begin{bmatrix} \gamma_k S_k^T S_k & L_k \\ L_k^T & -D_k \end{bmatrix}^{-1}, \ \widehat{Q} = RQ^T,$$

$$S_k = \begin{bmatrix} \boldsymbol{s}_{k-m} & \cdots & \boldsymbol{s}_{k-1} \end{bmatrix}, \ Y_k = \begin{bmatrix} \boldsymbol{y}_{k-m} & \cdots & \boldsymbol{y}_{k-1} \end{bmatrix},$$

$$\boldsymbol{s}_k = \boldsymbol{w}_{k+1} - \boldsymbol{w}_k, \ \boldsymbol{y}_k = \boldsymbol{g}_{k+1} - \boldsymbol{g}_k, \ \gamma_k = \boldsymbol{y}_{k-1}^T \boldsymbol{y}_{k-1} / \boldsymbol{s}_{k-1}^T \boldsymbol{y}_{k-1},$$

$$D_k = \mathrm{diag} \begin{bmatrix} \boldsymbol{s}_{k-m}^T \boldsymbol{y}_{k-m} & \cdots & \boldsymbol{s}_{k-1}^T \boldsymbol{y}_{k-1} \end{bmatrix}, \ (L_k)_{ij} = \begin{cases} \boldsymbol{s}_{k-m+i-1}^T \boldsymbol{y}_{k-m+j-1} & \text{if } i > j \\ 0 & \text{otherwise} \end{cases}.$$

The low rank structure of $B_k$ enables fast computation of subsequent coordinate descent procedure. Specifically, notice that all $Q, R, \widehat{Q}$, and $\mathrm{diag}(B)$ can be precomputed in $O(m^2 p + m^3)$ time, which is significantly smaller than naive Hessian inversion $O(p^3)$. After precomputation, in each coordinate update, both $a$ and $c$ in (17) can be computed and updated in $O(1)$ time. Moreover, let $\widehat{\boldsymbol{d}} = \widehat{Q}\boldsymbol{d} \in \mathbb{R}^{2m}$, we have $(B\boldsymbol{d})_j = \gamma \boldsymbol{d}_j - Q_{j,:}\widehat{\boldsymbol{d}}$, which suggests $b$ in (17) only requires $O(m)$ to compute and update. Therefore each coordinate update is $O(m)$.

The detailed procedure of PQN is outlined in Algorithm 2.

## C  Sensitivity of threshold

We demonstrate the effect of threshold in Figure 4. For each setting, we computed the "ROC" curve for FDR and TPR with varying level of threshold, while ensuring the resulting graph is indeed a DAG. On the right, we also present the estimated edge weights of $\widetilde{W}_{\mathsf{ECP}}$ in decreasing order. One can first observe that in all cases most of the edge weights are equal or close to zero as expected. The remaining question is how to choose a threshold that separates out these (near zero) from signals (away from zero) so that best performance can be achieved. With enough samples, one can often notice a sudden change in the weight distribution as in Figure 4(a)(c). With insufficient samples, the breakpoint is less clear, and the optimal choice that balances between TPR and FDR is depends on the specific settings. Nonetheless, the predictive performance is less sensitive to threshold value as one can see from the slope of the decrease in the weights before getting close to zero. Indeed, in our experiments, we found a fixed threshold $\omega = 0.3$ is a suboptimal yet reasonable choice across many different settings.

**Algorithm 2** Proximal Quasi-Newton for unconstrained problem [50]

1. Input: $\boldsymbol{w}_0, \boldsymbol{g}_0 = \nabla f(\boldsymbol{w}_0)$, active set $\mathcal{S} = [p]$.
2. For $k = 0, 1, 2, \ldots$:
    (a) Shrink $\mathcal{S}$ to rule out $j$ with $w_j = 0$ or small subgradient $|\partial_j L(\boldsymbol{w})|$
    (b) If shrinking stopping criteria is satisfied
        i. Reset $\mathcal{S} = [p]$ and L-BFGS memory
        ii. Update shrinking stopping criteria and continue
    (c) Solve (16) for descent direction $\boldsymbol{d}_k$ using coordinate update (17) on active set
    (d) Line search for step size $\eta \in (0, 1]$ until Armijo rule is satisfied:

$$f(\boldsymbol{w}_k + \eta \boldsymbol{d}_k) \leq f(\boldsymbol{w}_k) + \eta c_1 (\lambda \|\boldsymbol{w}_k + \boldsymbol{d}_k\|_1 - \lambda \|\boldsymbol{w}_k\| + \boldsymbol{g}_k^T \boldsymbol{d}_k), \quad (23)$$

   where $c_1$ is some small constant, typically set to $10^{-3}$ or $10^{-4}$.
    (e) Generate new iterate $\boldsymbol{w}_{k+1} \leftarrow \boldsymbol{w}_k + \eta \boldsymbol{d}_k$
    (f) Update $\boldsymbol{g}, \boldsymbol{s}, \boldsymbol{y}, Q, R, \widehat{Q}$ restricted to $\mathcal{S}$

(a) ER2, $n = 1000$              (b) ER2, $n = 20$

(c) SF4, $n = 1000$              (d) SF4, $n = 20$

Figure 4: Illustration of the effect of the threshold with $d = 20$ and $\lambda = 0.1$. For each subfigure, ROC curve (left) shows FDR and TPR with varying level of threshold, and sorted weights (right) plots the entries of $\widetilde{W}_{\text{ECP}}$ in decreasing order.

## D  Sensitivity of weight scale

We investigate the effect of weight scaling to the NOTEARS algorithm in Figure 5. In particular, we run experiments with $w_{ij} \in \alpha \cdot [0.5, 2] \cup -\alpha \cdot [0.5, 2]$ with $\alpha \in \{1.0, 0.9, 0.8, \ldots, 0.1\}$. On the left, we plot the smallest threshold $\omega$ required to obtain a DAG (see Section 4.3) for different scale $\alpha$. Overall, across different values of $\alpha$, the variation in the smallest $\omega$ required is minimal. We also hasten to point out that this also decreases the signal to noise ratio (SNR), which more directly affects the accuracy. Indeed, in the figure on the right, we can observe (as expected) some performance drop when using smaller value of $\alpha$.

Figure 5: Varying weight scale $\alpha \in \{1.0, \ldots, 0.1\}$ with $d = 20$ and $n = 1000$ on an ER-2 graph. (Left) Smallest threshold $\omega$ such that $\widehat{W}$ is a DAG. (Right) SHD between ground truth and NOTEARS, lower the better. The minimum $\omega$ remains stable, while the accuracy of NOTEARS drops as expected since the SNR decreases with $\alpha$.

## E  Experiments

### E.1  Experiment details

We used simulated graphs from two well-known ensembles of random graphs:

- *Erdös-Rényi (ER).* Random graphs whose edges are added independently with equal probability $p$. We simulated models with $d$, $2d$, and $4d$ edges (in expectation) each, denoted by ER-1, ER-2, and ER-4, respectively.

- *Scale-free networks (SF).* Networks simulated according to the preferential attachment process described in Barabási & Albert [5]. We simulated scale-free networks with $4d$ edges and $\beta = 1$, where $\beta$ is the exponent used in the preferential attachment process.

Scale-free graphs are popular since they exhibit topological properties similar to real-world networks such as gene networks, social networks, and the internet. Given a random acyclic graph $B \in \{0,1\}^{d \times d}$ from one of these two ensembles, we assigned edge weights independently from Unif $([-2, -0.5] \cup [0.5, 2])$ to obtain a weight matrix $W = [w_1 \mid \cdots \mid w_d] \in \mathbb{R}^{d \times d}$. Given $W$, we sampled $X = W^T X + z \in \mathbb{R}^d$ according to the following three noise models:

- *Gaussian noise* (Gauss). $z \sim \mathcal{N}(0, I_{d \times d})$.
- *Exponential noise* (Exp). $z_j \sim \mathrm{Exp}(1)$, $j = 1, \ldots, d$.
- *Gumbel noise* (Gumbel). $z_j \sim \mathrm{Gumbel}(0, 1)$, $j = 1, \ldots, d$.

Based on these models, we generated random datasets $\mathbf{X} \in \mathbb{R}^{n \times d}$ by generating the rows i.i.d. according to one of the models above. For each simulation, we generated $n$ samples for graphs with $d \in \{10, 20, 50, 100\}$ nodes. To study both high- and low-dimensional settings, we used $n \in \{20, 1000\}$.

For each dataset, we ran FGS, PC, and LinGAM and NOTEARS to compare the performance in reconstructing the DAG $B$. We used the following implementations:

- FGS and PC were implemented through the `py-causal` package, available at `https://github.com/bd2kccd/py-causal`. Both of these methods are written in highly optimized Java code.

- LinGAM was implemented using the author's Python code: `https://sites.google.com/site/sshimizu06/lingam`.

Since the accuracy of PC and LiNGAM was significantly lower than either FGS or NOTEARS, we only report the results against FGS. A few comments on FGS are in order: 1) FGS estimates a graph, so it does not output any parameter estimates; 2) Instead of returning a DAG, FGS returns a CPDAG [9], which contains undirected edges; 3) FGS has a single tuning parameter that controls the strength

|              |                         |                         |
|:------------:|:-----------------------:|:-----------------------:|
| (a) true graph | (b) estimate with $n = 1000$ | (c) estimate with $n = 20$ |

Figure 6: Visual comparison of the learned weighted adjacency matrix on a 20-node graph with $n = 1000$ (large samples) and $n = 20$ (insufficient samples): $\widetilde{W}_{\text{ECP}}(\lambda)$ is the proposed NOTEARS algorithm with $\ell_1$-regularization $\lambda$, and $B_{\text{FGS}}$ is the binary estimate of the baseline [31]. Top row: ER1, bottom row: ER4.

of regularization. Thus, in our evaluations, we treated FGS favourably by treating undirected edges as true positives as long as the true graph had a directed edge in place of the undirected edge. For tuning parameters, we used the values suggested by the authors of the FGS code.

Denote the estimate returned by FGS by $B_{\text{FGS}}$. As discussed in Appendix C, we fix the threshold at $\omega = 0.3$. Having fixed $\omega$, when there is no regularization, NOTEARS requires no tuning. With $\ell_1$-regularization, NOTEARS-$\ell_1$ requires a choice of $\lambda$ which wes selected as follows: Based on the estimate returned by FGS, we tuned $\lambda$ so that the selected graph (after thresholding) had the same number of edges as $B_{\text{FGS}}$ (or as close as possible). This ensures that the results are not influenced by hyperparameter tuning, and fairly compares each method on graphs of roughly the same complexity. Denote this estimate by $\widehat{W}$ and the resulting adjacency matrix by $\widehat{B} = \mathcal{A}(\widehat{W})$.

## E.2 Metrics

We evaluated the learned graphs on four common graph metrics: 1) False discovery rate (FDR), 2) True positive rate (TPR), 3) False positive rate (FPR), and 4) Structural Hamming distance (SHD). Recall that SHD is the total number of edge additions, deletions, and reversals needed to convert the estimated DAG into the true DAG. Since we consider directed graphs, a distinction between True Positives (TP) and Reversed edges (R) is needed: the former is estimated with correct direction whereas the latter is not. Likewise, a False Positive (FP) is an edge that is not in the undirected skeleton of the true graph. In addition, Positive (P) is the set of estimated edges, True (T) is the set of true edges, False (F) is the set of non-edges in the ground truth graph. Finally, let (E) be the extra edges from the skeleton, (M) be the missing edges from the skeleton. The four metrics are then given by:

1. FDR $= (R + FP)/P$
2. TPR $= TP/T$
3. FPR $= (R + FP)/F$
4. SHD $= E + M + R$.

## E.3 Further evaluations

Figure 6 shows learned weighted adjacency matrices for ER1 and ER4. One can observe the same trend: with large $n$, both regularized and unregularized NOTEARS works well compared to FGS, and with small $n$, due to identifiability, the unregularized NOTEARS suffers significantly, yet with the help of $\ell_1$-regularization we can still accurately recover the true underlying graph.

Figure 7 and Figure 8 shows structure recovery results for $n = 1000$ and $n = 20$ for various random graphs and SEM noise types. Other than fixed $\omega$ as in the main paper, we also included the optimal

Figure 7: Structure recovery results for $n = 1000$. Lower is better, except for TPR (lower left), for which higher is better. Rows: random graph types, {ER,SF}-$k$ = {Erdös-Rényi, scale-free} graphs with $kd$ expected edges. Columns: noise types of SEM. Error bars represent standard errors over 10 simulations.

choice of thresholding, marked as "best". The trend is consistent with the main text: our method in general outperforms FGS, without tuning $\omega$ to the optimum for each setting.

Table 2 extends the global minimizer result for various random graph types. For each random graph and samples, we computed exact local scores as inputs to GOBNILP program, which finds the globally optimal structure for the given score. We can again observe that the difference between our estimate $\widehat{W}$ and global minimizer $W_{\mathsf{G}}$ is small across all cases.

Figure 8: Structure recovery results for $n = 20$. Lower is better, except for TPR (lower left), for which higher is better. Rows: random graph types, {ER,SF}-$k$ = {Erdös-Rényi, scale-free} graphs with $kd$ expected edges. Columns: noise types of SEM. Error bars represent standard errors over 10 simulations.

Table 2: Comparison of NOTEARS vs. globally optimal solution. $\Delta(W_{\mathsf{G}}, \widehat{W}) = F(W_{\mathsf{G}}) - F(\widehat{W})$.

| $n$ | $\lambda$ | Graph | $F(W)$ | $F(W_{\mathsf{G}})$ | $F(\widehat{W})$ | $F(\widetilde{W}_{\mathsf{ECP}})$ | $\Delta(W_{\mathsf{G}}, \widehat{W})$ | $\|\widehat{W} - W_{\mathsf{G}}\|$ | $\|W - W_{\mathsf{G}}\|$ |
|---|---|---|---|---|---|---|---|---|---|
| 20 | 0.00 | ER1 | 5.01 | 3.69 | 5.19 | 3.73 | -1.50 | 0.09 | 3.54 |
| 20 | 0.50 | ER1 | 12.43 | 9.90 | 10.69 | 9.88 | -0.78 | 0.11 | 2.76 |
| 1000 | 0.00 | ER1 | 4.96 | 4.93 | 4.97 | 4.92 | -0.04 | 0.03 | 0.35 |
| 1000 | 0.50 | ER1 | 12.37 | 10.53 | 11.01 | 10.58 | -0.48 | 0.11 | 2.47 |
| 20 | 0.00 | ER2 | 5.11 | 3.85 | 5.36 | 3.88 | -1.52 | 0.07 | 3.38 |
| 20 | 0.50 | ER2 | 16.04 | 12.81 | 13.49 | 12.90 | -0.68 | 0.12 | 3.15 |
| 1000 | 0.00 | ER2 | 4.99 | 4.97 | 5.02 | 4.95 | -0.05 | 0.02 | 0.40 |
| 1000 | 0.50 | ER2 | 15.93 | 13.32 | 14.03 | 13.46 | -0.71 | 0.12 | 2.95 |
| 20 | 0.00 | ER4 | 4.76 | 3.66 | 5.23 | 3.88 | -1.57 | 0.08 | 4.25 |
| 20 | 0.50 | ER4 | 28.24 | 16.38 | 19.81 | 16.82 | -3.44 | 0.15 | 6.66 |
| 1000 | 0.00 | ER4 | 5.03 | 5.00 | 5.50 | 4.97 | -0.50 | 0.00 | 0.46 |
| 1000 | 0.50 | ER4 | 28.51 | 18.29 | 29.91 | 18.69 | -11.61 | 0.13 | 5.76 |
| 20 | 0.00 | SF4 | 4.99 | 3.77 | 4.70 | 3.85 | -0.93 | 0.08 | 3.31 |
| 20 | 0.50 | SF4 | 23.33 | 16.19 | 17.31 | 16.69 | -1.12 | 0.15 | 5.08 |
| 1000 | 0.00 | SF4 | 4.96 | 4.94 | 5.05 | 4.99 | -0.11 | 0.04 | 0.29 |
| 1000 | 0.50 | SF4 | 23.29 | 17.56 | 19.70 | 18.43 | -2.13 | 0.13 | 4.34 |