[Reviews · NeurIPS 2018]

Reviewer 1



The authors study the problem of structure learning for Bayesian networks. The conventional methods for this task include the constraint-based methods or the score-based methods which involve optimizing a discrete score function over the set of DAGs with a combinatorial constraint. Unlike the existing approaches, the authors propose formulating the problem as a continuous optimization problem over real matrices, which performs a global search, and can be solved using standard numerical algorithms. The main idea in this work is using a smooth function for expressing an equality constraint to force acyclicity on the estimated structure. The paper is very well written and enjoyable to read. I have the following comments: - Least-Squares loss is used in this work as the score function. It seems that the main reason for this choice is its desirability from the optimization point of view. I was wondering if there are any structure related guarantee when using this score function. For example, will the global optimum be in the Markov equivalence class of the ground truth DAG? Will we be able to detect the v-structures?, etc. - Two restricting assumptions in this work are linearity and the assumption of full observability. Both of these assumptions are perfectly reasonable for this stage of the research, but I was wondering if the authors have examined their approach on models which are violating these assumptions? Specifically, in the case of non-linear SEM, how would be the output related to the ground truth DAG? Or more importantly, how essential is additive noise for the performance of Least-Squares loss? - Augmented Lagrangian method is used for solving the equality-constrained program. My concern is that in this method, the penalty term may force some sparsity restrictions on the learned structure. This may prevent the proposed approach from working well for all desired levels of sparsity. - The main technical tool in this work is the use of function h(.) for characterizing acyclicity. This characterization seems to be new in the literature, but it would be a more clarifying if the authors mention that the connection between the trace of the powers of the adjacency matrix and cycles in the corresponding graph was known in the literature (for example, Frank Harary, Bennet Manvel "On the number of cycles in a graph," 1971). My concern about the function h(.) is regarding property (b) of function: "The values of h quantify the “DAG-ness” of the graph". I am not sure how true this claim is: First of all, the value of h depends on the edge weights in the graph. Therefore, two graphs with exactly same structure but different weights will get different values for h(W). This gets more problematic as we can design a graph with more cycles, but smaller weights and end up getting a lower value for h compared to another graph with fewer cycles but larger weights. - Another concern of mine regarding function h(.) is that it over counts the cycles: If there is just one loop (cycle of length 1) in the graph, its effect will appear in all powers of the adjacency matrix. Also, if there is a cycle of length 3 in the graph, it will cause a non-zero value for all of its three corresponding vertices on the diagonal of W^3. In line 422 in the supplementary materials, it is mentioned that tr(B+B^2+...) counts the number of cycles in B. Based on the aforementioned examples, can we say that this sentence is not necessarily true? A clarification in this regard would be appreciated. - I suspect that for small values of coefficients, specifically, if all were less than 1, for example, w_{ij}~unif([0.1 0.5]) the error of the proposed method will increase. This is due to the fact that h(w) will be small and the algorithm may make more errors. It would be appreciated if the authors also examine their proposed method on such values. - The simulations and experiments are sufficient and seem fair. -----After author feedback: I thank the authors for their responses. Based on the responses, I increase my score to 8. However, I still have concerns regarding the used score function, the properties of function h(.), and the performance on real data.

Reviewer 2



The paper presents a method for casting the problem of learning a Bayesian network that best explains observed data as a continuous optimisation problem on matrices. This has the advantage of allowing existing powerful solvers to be used. The presented results show an improvement in the quality of networks learned over a leading existing approach. The paper tackles an interesting and relevant question in which there has been a significant amount of research and improvement in recent years. The approach presented is motivated strongly in the paper by the desire to get away from discrete superexponential acyclicity constraints and replace them by a continuous function. In doing so, it creates a significantly different approach from existing work and could well lead to further work building on that presented here. The paper would benefit from it being made much clearer and earlier that the presented technique is a heuristic method rather than an exact global optimal method. The paper originally seems to promise a real breakthrough in the harder global optimal problem which it does not provide. This aside, the introductory sections of the paper provide a clear and precise overview of the material presented. The literature review mentions the relevant work that one would expect, but doesn't really get into any depth - for example it does not explain how acyclicity is handed in the approaches it will go on to compare to. The description of the novel contribution and the methods used to solve the problem are clear. I can follow the techniques used but lack sufficient knowledge to say whether these are the most suitable for this problem. The experiments performed are valid ways to compare the methods and do not appear to bias in favour of the new algorithm. The results show that the technique generally outperforms one of the top existing approaches. There are always more experiments that could be performed and further ways in which data could be analysed but the presented results are generally sufficient for the conference. One notable missing item from the results is any mention of runtime. This is (almost) as important as solution quality when considering heuristic algorithms. It would also be beneficial to see the results obtained on some standard benchmarks in order to more easily compare with other past and future techniques. Overall, the technique is an interesting and very novel approach to an important problem, with results that appear significant. There is a great deal of clarity throughout the paper, though more attention could be paid to situating the approach within the existing field.

Reviewer 3



The paper proposes to learn the structure of Bayesian networks via continuous optimization, where the key contribution is to use the trace of the matrix exponential of the element-wise squared adjacency matrix as acyclicity measure, which can be exploited in numeric optimization. For optimization, a simple and effective augmented Lagrangian approach is proposed. Strengths: + While the used characterisation of acyclicity is widely known, to the best of my knowledge it has not been exploited for structure learning. Thus the paper is original. + the optimization scheme appears to be natural and effective. + The experimental evaluation shows the efficacy of the approach, delivering convincing results. Weaknesses: - the d^3 computational cost hinders scalability of the approach, and will likely squeeze a tear out of a practitioner's eye, when applying the method to a large problem. - it is not clear, whether the method could be applied to more classical cost functions, e.g. BDe. Quality: The key contribution is clearly the continuous characterization of DAG acyclicity via the matrix exponential, which is a very natural approach. It is actually surprising that this has not been done before, so that this paper fills a gap with a convincing approach. One reason why this approach has not been published before might be certain practical difficulties concerning optimization, when using the matrix exponential. Perhaps the authors could comment on this, in particular whether they have tried other, less successful approaches. Clarity: The paper is very clear and reproducible. Originality: While the characterization of acyclicity was definitely around in folk wisdom, it seems that this paper is the first to actually successfully use it. Thus, an original contribution. Significance: While, as mentioned above, the approach probably does not scale well (d^3 computational costs), the stage is open to improve on that with various approximations. It is highly likely that this work will stimulate further research in this direction. *** EDIT *** I read the authors' response and they have adequately addressed (my anyway minor) concerns.